# Phytochemicals in Chinese Chive (*Allium tuberosum*) Induce the Skeletal Muscle Cell Proliferation via PI3K/Akt/mTOR and Smad Pathways in C2C12 Cells

**DOI:** 10.3390/ijms22052296

**Published:** 2021-02-25

**Authors:** Mira Oh, Seo-Young Kim, SeonJu Park, Kil-Nam Kim, Seung Hyun Kim

**Affiliations:** 1College of pharmacy, Yonsei Institute of Pharmaceutical Sciences, Yonsei University, Incheon 21983, Korea; purunmei2002@naver.com; 2Chuncheon Center, Korea Basic Science Institute (KBSI), Chuncheon 24341, Korea; kimsy11@kbsi.re.kr (S.-Y.K.); sjp19@kbsi.re.kr (S.P.)

**Keywords:** *Allium tuberosum*, Chinese chive, cell proliferation, flavonoid glycoside, skeletal muscle differentiation

## Abstract

Chinese chive (*Allium tuberosum*) is a medicinal food that is cultivated and consumed mainly in Asian countries. Its various phytochemicals and physiological effects have been reported, but only a few phytochemicals are available for skeletal muscle cell proliferation. Herein, we isolated a new compound, kaempferol-3-*O*-(6″-feruloyl)-sophoroside (**1**), along with one known flavonoid glycoside (**2**) and six amino acid (**3**–**8**) compounds from the water-soluble fraction of the shoot of the Chinese chive. The isolated compounds were identified using extensive spectroscopic methods, including 1D and 2D NMR, and evaluated for their proliferation activity on skeletal muscle cells. Among the tested compounds, newly isolated flavonoid (**1**) and 5-aminouridine (**7**) up-regulated PI3K/Akt/mTOR pathways, which implies a positive effect on skeletal muscle growth and differentiation. In particular, compound **1** down-regulated the Smad pathways, which are negative regulators of skeletal muscle growth. Collectively, we suggest that major constituents of Chinese chive, flavonoids and amino acids, might be used in dietary supplements that aid skeletal muscle growth.

## 1. Introduction

There is an age-related loss in skeletal muscle mass and strength, known as sarcopenia [1]. Sarcopenia poses a huge health care burden in the elderly [2]. Therefore, many researchers are interested in finding dietary ingredients from natural products that can help increase and maintain skeletal muscle mass.

Chinese chive (*Allium tuberosum*, Liliaceae) is a hardy perennial plant, which has edible thin leaves. It is mainly cultivated and consumed in east and southeast Asia not only as food but also as medicine. According to the dictionary of Chinese drugs [3], Chinese chive has been used for the treatment of abdominal pain, diarrhea, hematemesis, snakebite, and asthma. Indeed, several studies of Chinese chive’s biological and physiological activities support its wide range of medicinal effects; Tang et al. (2017) [4] confirmed the antidiabetic and hepatoprotective effect of Chinese chive extract’s butanol fraction, and they expected that it might be due to the influence of protective effect against liver injury. Chinese chive also showed significant hypolipidemic activity in hyperlipidemic guineapigs by reducing serum cholesterol, triglyceride, LDL-C, and atherogenic index [5]. The report about cytotoxic activity indicated that thiosulfinates from Chinese chive inhibited cell proliferation in HepG2 via apoptosis [6]. Despite these diverse physiological activities, there are no reports about skeletal muscle cell proliferation activity of Chinese chive.

However, muscle-related reports have been published using *Allium* genus. For instance, Gautam et al. (2015) [7] confirmed that onion (*A. cepa*) stimulates glucose uptake, which is related to skeletal muscle growth and mass increase in L6 rat skeletal muscle cells [8]. In addition, macrostemonoside A, a new compound isolated from *A. macrostemon* Bung, promoted energy metabolism in muscles during its action mechanisms on mice [9]. More recently, Kalhotra et al. (2020) [10] reported that garlic bulb (*A. sativum*) extract on L6 cells increased cell proliferation and/or differentiated skeletal muscle cells compared to untreated cells.

In this study, we isolated and identified phytochemicals from the methanol extracts of Chinese chive to find a substance that has a positive effect on muscle growth. Using various chromatographic techniques and extensive spectroscopic methods, eight compounds were identified. Among tested compounds, a new flavonol glycoside and an amino acid exhibited significant skeletal muscle cell proliferation.

## 2. Results and Discussion

### 2.1. Chemical Composition of Chinese Chive

LC–MS total ion current (TIC) chromatograms were acquired from the Chinese chive’s shoots sample for spectrometric analysis of compound composition. Methanol extract of the sample was strongly ionized in the ESI-negative mode (Figure 1).

In order to identify the various compounds of Chinese chive, molecular networking was performed. Recently, network annotation propagation (NAP) was introduced as an in silico annotation tool, which predicts the compound structure of unknown fragmented mass spectrum with a re-ranking system to increase annotation accuracy [11]. Molecular networking of Chinese chive was generated using the raw negative ion mode LC–MS data and visualized through Cytoscape 3.8.0, an open-source software for visualizing complex networks [12] (Appendix A). Of the 102 nodes, the structures of 71 nodes were predicted in silico. The molecular networking work on NAP can be found at Center for Computational Mass Spectrometry. Available online: https://proteomics2.ucsd.edu/ProteoSAFe/status.jsp?task=3f047829fb864932b2e4e49c656b1920 (accessed on 20 December 2020).

Chemical classification of 66 nodes, of 71 nodes annotated with NAP, was yielded through ClassyFire program [13]. The “phenylpropanoids and polyketides” group showed the largest proportion (22.7%) among the seven superclass levels, followed by “lipid and lipid-like molecules” (21.2%) and “organoheterocyclic compounds” (18.2%). The largest chemical constituents that included in the “phenylpropanoids and polyketides” group were “flavonoids” (Figure 2). Of the 26 class levels detected in Chinese chive, the following seven classes constituted the top 54.5%; “steroids and steroid derivatives”, “lupin alkaloids”, “benzene and substituted derivatives”, “carboxylic acids and derivatives”, “glycerophospholipids”, “flavonoids”, and “piperidines”. The subclass levels constituting each of the seven classes are shown in Figure 3.

### 2.2. Isolation and Identification of Phytochemicals in Chinese Chive Extract

The methanol extract of the shoots of Chinese chive was suspended in water and partitioned using CHCl_3_ and EtOAc to obtain three layers. By means of various chromatographic and isolation approaches, one new compound along with eight known compounds were isolated (Figure 4). The known compounds were compared with the reported ^1^H, ^13^C NMR, and MS data and were identified as kaempferol-3-*O*-sophoroside (**2**) [14], thymidine (**3**) [15], 2’-deoxyadenosine and adenosine (**4, 5**) [16], uridine (**6**) [17], 5-aminouridine (**7**), and tryptophan (**8**) [18].

Compound **1** was obtained as a yellow amorphous powder, and its molecular formula was determined as C_37_H_38_O_19_ by HR-ESI-MS ion at *m*/*z* 809.1889 [M + Na]^+^ (calcd for C_37_H_38_O_19_, 809.1900) (Appendix A). The ^1^H-NMR spectrum exhibited the existence of kaempferol backbone: A_2_B_2_-type benzene ring at *δ*_H_ 6.90 (2H, d, *J* = 8.6 Hz) and 8.06 (2H, d, *J* = 8.6 Hz) and two methine signals at *δ*_H_ 6.12 (d, *J* = 2.1 Hz) and 6.18 (d, *J* = 2.1 Hz). In addition, ^1^H-NMR spectrum exhibited the signals of feruloyl moiety: two olefinic methine signals at *δ*_H_ 6.05 (d, *J* = 15.9 Hz) and 7.34 (d, *J* = 15.9 Hz) and ABC-type benzene ring at *δ*_H_ 6.65 (d, *J* = 8.1 Hz), 6.72 (dd, *J* = 1.9, 8.2 Hz), and 6.82 (d, *J* = 1.9 Hz), as well as methoxy proton at *δ*_H_ 3.79 (s). The ^13^C NMR spectrum of **1** showed the presence of a total of 37 carbons, 13 quaternary carbons (1 carboxyl, 1 carbonyl, 8 oxygenated), 21 methines (ten oxygenated), 2 oxygenated methylenes, and 1 methoxy carbon (Table 1). The NMR data of **1** were similar to those of kaempferol 3-*O*-[2-*O*-(β-glucopyranosyl)-6-*O*-(*trans*-sinapoyl)-β-glucopyranoside] [19] except for the replacement of sinapoyl into feruloyl moiety. While the sinapoyl moiety normally exhibits A_2_-type benzene ring with two methoxy groups, the NMR spectrum of compound **1** showed the ^1^H-NMR signals of ABC-type benzene ring containing one methoxy group, which was assigned as a feruloyl moiety. Two olefinic protons at H- α (*δ*_H_ 7.34) and H- β (*δ*_H_ 6.05) of the feruloyl moiety with large coupling constants (*J* = 15.9 Hz) indicated that the geometry of the olefin was *trans*-configuration. Both sugar moieties were identified as β-D-glucopyranoside (glucoses A and B) by comparing with the sugar moieties in quercetin-3-*O*-(6″-feruloyl)-sophoroside (Table 1) [20] and acid hydrolysis of **1** (see experimental). The HMBC correlation between the methoxy proton (*δ*_H_ 3.79) and the carbon signals at C-3′′′′ (*δ*_C_ 149.2) confirmed the location of the methoxy at C-3′′′′ (Figure 5). The location of sugar moieties at C-3 was assigned by the HMBC correlation from H-1″ (*δ*_H_ 5.12) to C-3 (*δ*_C_ 135.3), while the location of glucose B was designated at C-2″ by the HMBC correlation between H-1′′′ (*δ*_H_ 4.75) and C-2″ (*δ*_C_ 84.7). Furthermore, the position of feruloyl moiety at C-6″ was confirmed by the HMBC correlation between the H-6″ (*δ*_H_ 4.44 and 4.45) and C-6″ (*δ*_C_ 169.0) (Figure 5). Based on the above data, the structure of compound **1** was determined as kaempferol-3-*O*-(6″-feruloyl)-sophoroside. Raw 1D and 2D NMR spectrum data are shown in supporting information (Appendix A–S6).

### 2.3. Skeletal Muscle Cell Proliferation Activities of Phytochemicals from Chinese Chive on C2C12 Cells

Prior to evaluating the cell proliferation effect of isolated compounds from Chinese chive, their viability on C2C12 myoblasts was measured by 3-(4-5-dimethyl-2yl)-2-5-diphynyltetrasolium bromide (MTT) assay. All compounds did not show a cytotoxic effect on C2C12 cells at the concentration of 50 μM except for compound **4** (Figure 6A).

The skeletal muscle cell proliferation of isolated compounds was assessed using 5-bromo-2′-deoxyuridine (BrdU) cell proliferation assay. As shown in Figure 6B, compounds **1**, **2**, and **7** significantly increased the proliferation of myoblast compared to the non-treated cells. However, the other five compounds did not exert any effect on cell proliferation. Among them, compounds **3**–**5** significantly reduced cell proliferation. This result may be caused by cell toxicity due to three treatments of the compounds during the cell differentiation process. Therefore, we selected compound **1**, which is a newly isolated compound from Chinese chive in the present study, and compound **7**, which displayed the highest cell proliferation efficacy among three proliferative compounds, and evaluated the effect of skeletal muscle growth regulation.

### 2.4. Effect of Skeletal Muscle Growth Regulation of Compounds ***1*** and ***7*** from Chinese Chive via PI3K/Akt/mTOR and/or Smad Signaling Pathways

The differentiation and maturation of skeletal muscle require interactions between signaling pathways [21]. For instance, insulin-like growth factor-1 (IGF-1) plays a crucial role in the control of skeletal muscle growth during development [22]. IGF-1 binding to its receptor induces a phosphorylation of phosphoinositide 3 kinase (PI3K)/Akt pathway, resulting in myoblast differentiation and survival [18]. In addition, Akt activates mammalian target of rapamycin (mTOR) and increases protein synthesis via activation of ribosomal protein S6 kinase (S6K) [23]. Therefore, we analyzed the expression of proteins in the signaling pathways discussed below through Western blot analysis to determine whether two active compounds (**1** and **7**) regulate signaling pathways. 

As shown in Figure 7B–D, compound **1** significantly increased the protein expression levels of p-PI3K, p-Akt, and p-FoxO1 compared to the non-treated cells. Compound **7** also significantly promoted these protein expression levels comparing the non-treated cells (Figure 8B–D). Then, we analyzed the expression level of mTOR pathways, which are involved in cell proliferation and differentiation by p-Akt. The protein expression level of p-mTOR and p-70S6K were significantly increased by both compounds **1** and **7** compared to the non-treated cells (Figure 7E–F and Figure 8E–F). However, these protein expression levels were more activated in cells with the treatment of compound **7** than that of **1**. Collectively, both compounds induce muscle cell growth through p-mTOR/p70S6K via the p-PI3K/Akt, and compound **7** is especially believed to activate this signaling pathway more than compound **1** (Appendix A). We moved further to assess the expression level of MyoD, a protein in animals that plays a major role in regulating muscle development and takes part in the repair of damaged skeletal muscle by PI3K/Akt/mTOR signaling pathways [22,24]. As shown in 7G and 8G, the protein expression level of MyoD was significantly increased by both compounds **1** and **7**. The expression level of MyoD protein was also higher in the cells treated with compound **7** than that of **1** (Appendix A). In addition, it is believed to be due to the results that the expression level of PI3K/Akt/mTOR was more activated by treating compound **7** than that of compound **1**. On the other hand, in cells treated with a high concentration of compound **7**, the expression level of MyoD was not different compared to that of the non-treated cells. Flavonoids and plant extracts containing flavonoid derivatives have been reported to activate osteogenic and myogenic differentiation as well as glucose resistance regulation via PI3K/Akt/mTOR pathways [25,26,27,28]. Meanwhile, amino acids are important regulators of the activation of mTOR complex 1, which regulates essential cellular processes such as growth, proliferation, and survival [29]. Furthermore, amino acids regulate cell proliferation by regulating PI3K/Akt signaling and module mTOR complexes [30,31]. In accordance with previous studies, amino acids also positively regulate the PI3K/Akt/mTOR pathways in C2C12 cells. Based on the present studies, we suggest that two active compounds obtained from Chinese chive affect myogenic differentiation via PI3K/Akt/mTOR pathways in C2C12 cells.

The other pathway involved in muscle growth and differentiation is the Smad signaling pathways that are activated in response to TGF-β. Activated TGF-β receptors phosphorylate Smad2/3, and each of the two receptor-activated Smads forms heteromeric complexes with Smad 4 [32]. This complex translocates to the nucleus and down-regulates transcription of target genes such as MyoD. Therefore, we tried to determine whether compounds **1** and **7** down-regulate these Smad pathways. As shown in Figure 9, compound **1** significantly decreased the protein expression levels of p-Smad2/3 and Smad4 compared to the non-treated cells. Similarly, several studies were reported that flavonoid found in plants down-regulate the Smad signaling pathways [33,34,35]. However, in compound **7**-treated cells, both p-Smad2/3 and Smad4 protein did not decrease but rather increased their expression levels (Figure 10).

Overall, compound **1** not only activated the PI3K/Akt/mTOR signaling pathways but also down-regulated the Smad pathways. On the other hand, compound **7**, an amino acid, promoted only the PI3K/Akt/mTOR signaling pathways but could not regulate the Smad pathways. Therefore, it can be deduced that compound **1** exerted more influence on the expression of MyoD, a factor in the nucleus involved in skeletal muscle development, than that of compound **7**. Therefore, our results suggest that newly isolated flavonol glycoside and 5-aminouridine have skeletal muscle growth and development effects by promoting the PI3K/Akt/mTOR pathways and down-regulating Smad pathways (Figure 11).

## 3. Materials and Methods

### 3.1. General Experimental Procedures

Chemical shifts are reported in parts per million from TMS. All NMR spectra were recorded on an Agilent 400-MR-NMR spectrometer operated at 400 and 100 MHz for hydrogen and carbon, respectively. Data processing was carried out with the MestReNova ver.6.0.2 program. The IR spectra were obtained from a Tensor 37 FT-IR spectrometer (Bruker, Ettlingen, Germany). UV and Circular dichroism spectrums were determined on a Chirascan™ CD spectrometer. Preparative HPLC was carried out using an AGILENT 1200 HPLC system. Column chromatography (CC) was performed on silica-gel (Kieselgel 60, 70–230 mesh and 230–400 mesh, Merck, Kenilworth, NJ, USA), YMC RP-18 resins (30–50 μm, Fujisilisa Chemical Ltd., Kasugai, Aichi, Japan), or Diaion HP-20 regins (200–300 mesh, Mitsubishi Chemical Co., Chiyoda, Tokyo, Japan). For thin-layer chromatography (TLC), pre-coated silica-gel 60 F254 (0.25 mm, Merck) and RP-18 F254S (0.25 mm, Merck) plates were used.

### 3.2. Plant Material

Shoots of Chinese chive were collected at the Naju, Jeolla Province, South Korea in July 2018. A voucher specimen (AT201807) is deposited at the Herbarium of College of Pharmacy, Yonsei Institute of Pharmaceutical Sciences, Yonsei University, Incheon, Korea.

### 3.3. UPLC-QTOF-MS Analysis

Methanol extracts of Chinese chive’s shoots (1 mg/mL, 10 μL) was analyzed by the instrument consisted of a Waters ACQUITY UPLC system (Waters Corp., Milford, MA, USA) connecting to a quadrupole time of flight mass spectrometer (Xevo G2-XS QTOF, Waters Corp.). YMC-Triart C18 column (2.0 × 150 mm, 1.9 μm; YMC KOREA Co., Seongnam, Korea) was used for compound separation at the condition of 25 °C. The mobile phases were water (0.1% formic acid) and ACN (0.1% formic acid) with the following gradients: 5–95% ACN (0–20 min), 95–100% ACN (20–20.1 min), and 100% ACN (20.1–25 min). The flow rate was 4 mL/min.

### 3.4. Extraction and Isolation

Shoots of Chinese chive (4.0 kg) were dried at 40 °C for one day, and the final dry mass was 450 g. Dried shoots were extracted with MeOH under sonication at 30 °C to yield an extract (83.7 g), which was then suspended in H_2_O and successively partitioned using hexane, CHCl_3,_ and EtOAc to obtain EtOAc (AT-1C, 3.22 g) and H_2_O (AT-1D, 71.5 g) extracts after removal of the solvents in vacuo. The H_2_O fraction (AT-1D, 71.5 g) was subjected to a Diaion HP-20 column and eluted with 25, 50, and 75% MeOH to yield three sub-fractions, AT-W2A (3.8 g), AT-W2B (1.8 g), AT-W2C (1.1 g). The AT-W2B fraction was combined with AT-W2A fraction, and the integrated mass of AT-W2A (5.6 g) was applied to a silica gel column, and by eluting with CHCl_3_: MeOH: H_2_O (2:1:0.2, *v*/*v*/*v*) gave six sub-fractions, AT-W3A (0.2 g), AT-W3B (0.3 g), AT-W3C (0.4 g), AT-W3D (0.3 g), AT-W3E (0.3 g), and AT-W3F (0.5 g). The AT-W3A fraction was subjected to HPLC system, eluted with MeCN: H_2_O (7:93) to yield **3** (13.8 mg) and **4** (14.8 mg). Subsequently, AT-W3B and AT-W3C fractions were eluted with the same HPLC conditions and solvent composition as above to yield **5** (3.5 mg) and **6** (17.5 mg) respectively. Compound **7** (3.3 mg) and **8** (8.4 mg) were obtained from fraction AT-W3E, also eluted with MeCN: H_2_O (7:93). AT-W2C (1.1 g) was applied to a silica gel column, and by eluting with CHCl_3_: MeOH: H_2_O (2:1:0.2, *v*/*v*/*v*) gave four smaller fractions, AT-W4A (0.2 g), AT-W4B (0.1 g), AT-W4C (0.1 g), AT-W4D (0.2 g). The AT-W4B fraction was subjected to HPLC as above, eluted with MeCN: H_2_O (25:75) to yield **1** (10.0 mg), whereas the AT-W4C fraction gave **2** (15.1 mg) when eluted with the same solvent composition.

### 3.5. Kaempferol-3-O-(6ʺ-feruloyl)-sophoroside (***1***)

Yellow amorphous powder; C_37_H_38_O_19_, HR-ESI-MS *m*/*z*: 809.1889 [M + Na]^+^ (calcd for C_37_H_38_O_19_Na, 809.1900); for ^1^H (CD_3_OD, 400 MHz) and ^13^C NMR (CD_3_OD, 100 MHz) spectroscopic data; see Table 1.

### 3.6. Acid Hydrolysis of ***1***

Compound **1** (2 mg) was dissolved in 1 N HCl (dioxane–H2O, 1: 1, 1 mL) and heated to 80 °C in a water bath for 3 h. The acidic solution was neutralized with silver carbonate, and the solvent was thoroughly driven out under N_2_ gas overnight. After extraction with CHCl_3_, the aqueous layer was concentrated to dryness using N_2_ gas. The residue was dissolved in 0.1 mL of dry pyridine, followed by addition of L-cysteine methyl ester hydrochloride in pyridine (0.06 M, 0.1 mL). The reaction mixture was heated at 60 °C for 2 h. Trimethylsilylimidazole solution (0.1 mL) was then added, followed by heating at 60 °C for 1.5 h. The dried product was partitioned with n-hexane and water (0.1 mL each), and the organic layer was analyzed by gas chromatography (GC): column SPB-1 (0.25 mm × 30 m), detector FID, column temp 210 °C, injector temp 270 °C, detector temp 300 °C, carrier gas He (30 mL/min). Under these conditions, standard sugars gave peaks at t_R_ (min) 8.55 and 9.25 for D- and L-glucose, respectively. Peaks at t_R_ (min) 8.55 of D-glucose for **1** were observed.

### 3.7. Materials for Skeletal Muscle Cell Proliferation Activity

C2C12, a mouse myoblasts cell line, was purchased from the American Type Culture Collection (ATCC, Manassas, VA, USA). Dulbecco’s Modified Eagle Medium (DMEM), fetal bovine serum (FBS), horse serum (HS), penicillin-streptomycin, trypsin-EDTA, and Dulbecco’s Phosphate Buffered Saline (DPBS) were acquired from Gibco-BRL (Burlington, Ont, Canada). All the other chemicals used were of analytical grade.

### 3.8. Cell Culture and Differentiation

C2C12 cells were maintained in DMEM supplemented with 10% heat-inactivated FBS, streptomycin (100 mg/mL), and penicillin (100 unit/mL) at 37 °C under 5% CO_2_ humidified incubator. To induce differentiation, 80% confluent cultures were switched to DMEM containing 2% HS for six days with medium changes every other day.

### 3.9. Skeletal Muscle Cell Proliferation Activity

First, cytotoxicity was evaluated before measuring the cell proliferation activity of eight compounds isolated from Chinese chive. The cytotoxic assessment was performed according to the method described in Muthuramalingam et al. (2019) [36] with a slight modification. Briefly, after 24 h of cell seeding with 1 × 10^5^ cells per well (96 well plate), isolated compounds were treated for 24 h. Afterward, cytotoxic assessment was performed using MTT assay. The formazan crystals were dissolved in DMSO, and the absorbance was measured using an ELISA plate reader at 540 nm (BioTek Instruments, Inc., Winooski, VT, USA). The optical density of the formazan generated in non-treated control cells was considered to represent 100% viability. To assess the skeletal muscle cell proliferation activity, activities were determined using BrdU assay (Millipore, Billerica, MA, USA) according to the method used in Kim et al. (2019) [37]. Briefly, after 48 h of the initial cell seeding with 5 × 10^4^ cells per well (96 well plate), the growth medium (DMEM containing 10% FBS) was replaced with differentiation medium (DMEM containing 2% HS) and treated phytochemicals every other day during differentiation period. Finally, six days after the differentiation induction, cell proliferation activity was measured using BrdU assay.

### 3.10. Western Blot Analysis

Western blot analysis was carried out to observe the effect of eight compounds on skeletal muscle differentiation through the analysis of the expression of proteins involved in the myogenesis signaling cascade using the protocol described by Muthuramalingam and Kim et al. (2019). The eight compounds were treated with differentiation media (DMEM containing 2% HS) every each other during the cell differentiation period. Afterward, cytoplasmic and nuclear proteins were collected, and cellular proteins were separated by 12% sodium dodecyl sulfated (SDS)-polyacrylamide gel electrophoresis (PAGE) and transferred to nitrocellulose membranes. The membranes were blocked with 5% skim milk (in Tris buffered saline containing 0.2% Tween-20) and incubated overnight at 4 °C with relevant primary antibodies (1:1000 dilution in 5% skim milk) to identify proteins that inhibit or activate skeletal muscle cell synthesis. Following the HRP-conjugated secondary antibodies (1:3000 dilution in 5% skim milk), they were added to the membranes and were incubated for 90 min at room temperature. Finally, the bands were developed using chemiluminescent substrate and photographed using a FUSION FX Spectra program equipped with eVo-6 camera (Vilber Lourmat, Marne-La-Vallée, France). The band intensities were quantified using the Image J program.

## 4. Conclusions

In this study, we investigated the muscle growth activity of newly discovered flavonoid and amino acids from Chinese chive. Our results demonstrated that two active compounds interacted with muscle-related signaling pathways and induced skeletal muscle growth and differentiation. In particular, compound **1** promoted PI3K/Akt/mTOR pathways, which are positive signaling pathways, while simultaneously decreasing Smad pathways, the negative regulator of skeletal muscle growth. Compound **7** also activated PI3K/Akt/mTOR pathways. Taken together, we suggest that major constituents of Chinese chive, identified as flavonoids and amino acids, might be used as functional food candidates for aiding muscle growth and differentiation.

## Figures and Tables

**Figure 1 ijms-22-02296-f001:**
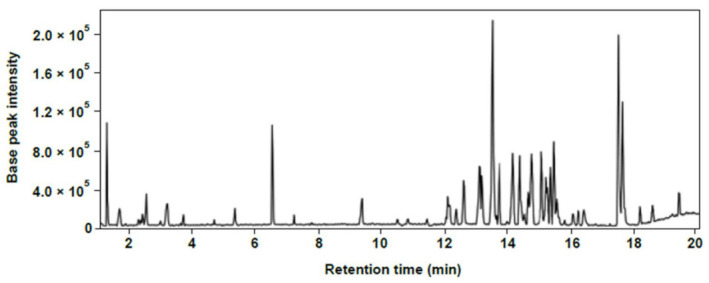
Representative LC-MS total ion current (TIC) chromatograms of methanol extract of Chinese chive in negative ion mode.

**Figure 2 ijms-22-02296-f002:**
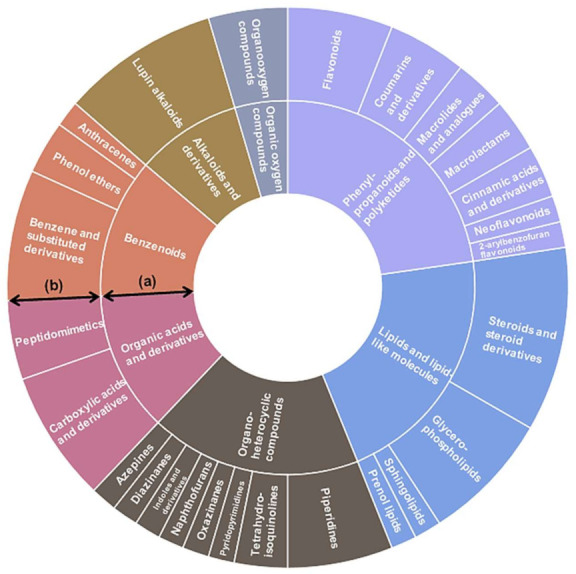
Sunburst plot of organic compounds annotated in Chinese chives. Nodes annotated by NAP were classified as different class levels using ClassyFire. (a) superclass level; (b) class level.

**Figure 3 ijms-22-02296-f003:**
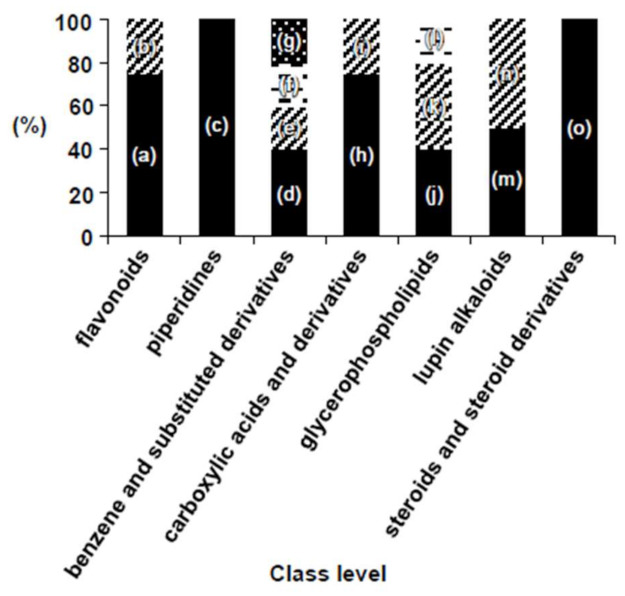
Bar plot of organic compounds annotated in Chinese chives. Each bars of seven class levels consist of one or more subclass levels. (a) flavonoid glycosides; (b) flavans; (c) phenylpiperidines; (d) benzyl alcohols; (e) benzoic acids and derivatives; (f) diphenylmethanes; (g) sulfanilides; (h) amino acids and derivatives; (i) peptides; (j) glycerophosphocholines; (k) glycerophosphoethanolamines; (l) glycerophosphoinositols; (m) cytisine and derivatives; (n) matrine alkaloids; (o) ergosterols and derivatives.

**Figure 4 ijms-22-02296-f004:**
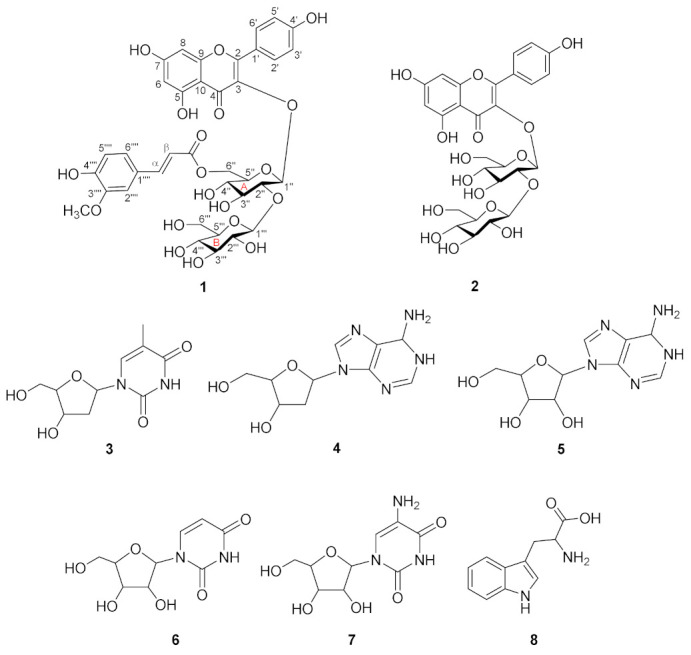
Chemical structures of compounds **1**−**8**. **1**, kaempferol-3-*O*-(6′′-feruloyl)-sophoroside; **2**, kaempferol-3-*O*-sophoroside; **3**, thymidine; **4**, 2′-deoxyadenosine; **5**, adenosine; **6**, uridine; **7**, 5-aminouridine; **8**, tryptophan.

**Figure 5 ijms-22-02296-f005:**
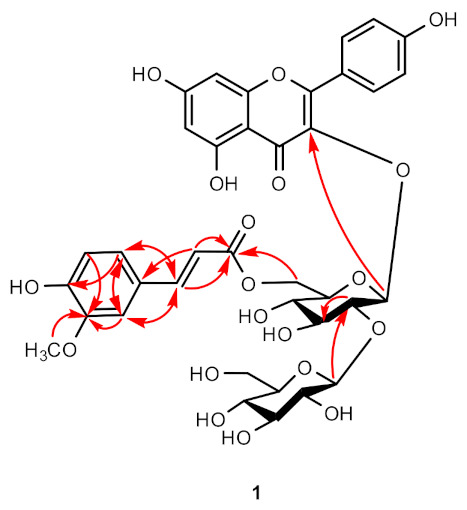
Key HMBC correlations of **1**.

**Figure 6 ijms-22-02296-f006:**
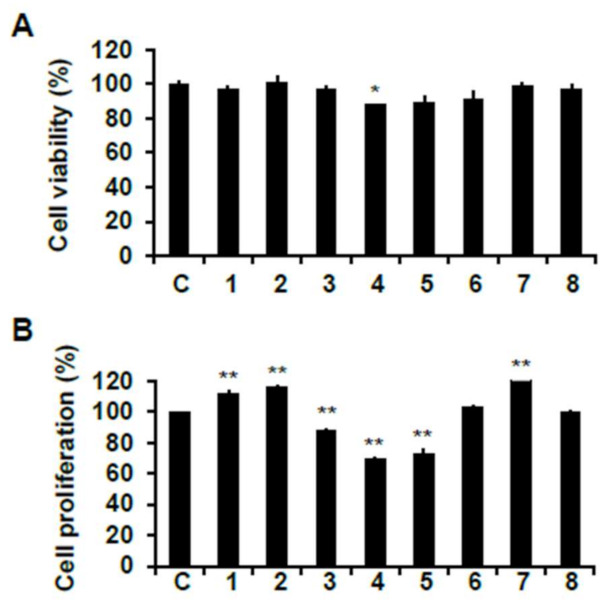
Skeletal muscle cell proliferation activities of phytochemicals from Chinese chive on C2C12 cells. (**A**) Cytotoxicity of phytochemicals on C2C12 cells. Cells were incubated for 24 h after seeding, and then treated phytochemicals for 24 h. After, MTT assay was performed. (**B**) Cell proliferation activity of phytochemicals on C2C12 cells. Cells were incubated for 48 h after seeding, and phytochemicals were treated while being replaced with three differentiation medium during six days of the differentiation period. After the differentiation period, cell proliferation activity was measured using BrdU assay. C, control. Experiments were performed in triplicate and the data were expressed as mean ± SE; * *p* < 0.01, ** *p* < 0.01 as compared to the control (untreated cells).

**Figure 7 ijms-22-02296-f007:**
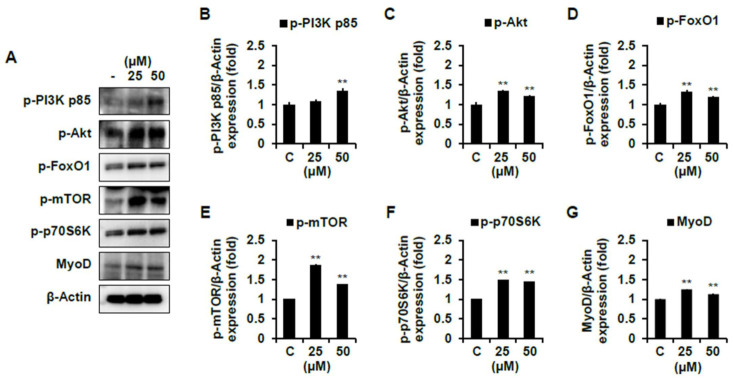
Skeletal muscle cell differentiation activity of compound **1** on C2C12 differentiated cells. The levels of p-PI3K p85, p-Akt, p-FoxO1, p-mTOR, p-p70S6K, and MyoD proteins were determined using a Western blot analysis. Every level of proteins versus β-Actin were measured by densitometry. C, control. Experiments were performed in triplicate and the data are expressed as mean ± SE; ** *p* < 0.01 as compared to the control (untreated cells).

**Figure 8 ijms-22-02296-f008:**
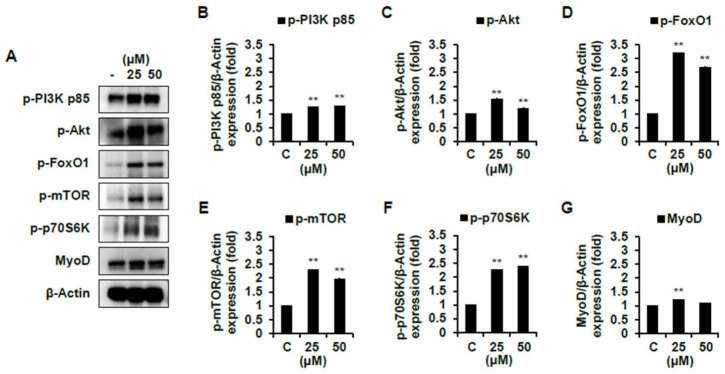
Skeletal muscle cell differentiation activity of compound **7** on C2C12 differentiated cells. The levels of p-PI3K p85, p-Akt, p-FoxO1, p-mTOR, p-p70S6K, and MyoD proteins were determined using a Western blot analysis. Every level of proteins versus β-Actin were measured by densitometry. C, control. Experiments were performed in triplicate and the data are expressed as mean ± SE; ** *p* < 0.01 as compared to the control (untreated cells).

**Figure 9 ijms-22-02296-f009:**
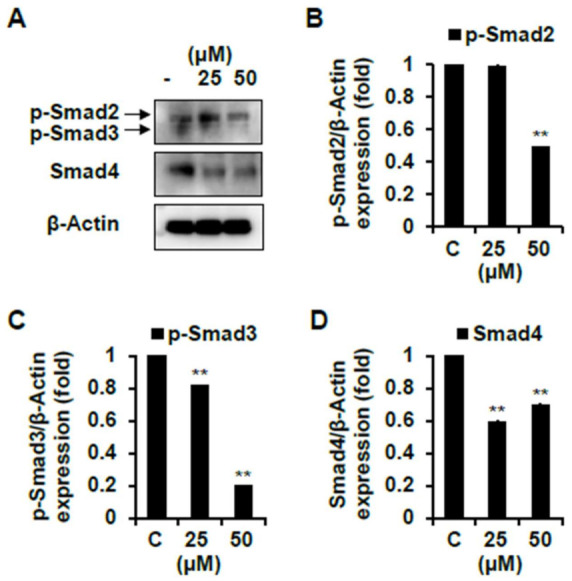
Skeletal muscle cell differentiation activity of compound **1** on C2C12 differentiated cells. The levels of p-Smad2/3 and Smad4 proteins were determined using a Western blot analysis. Every level of proteins versus β-Actin were measured by densitometry. C, control. Experiments were performed in triplicate and the data were expressed as mean ± SE; ** *p* < 0.01 as compared to the control (untreated cells**).**

**Figure 10 ijms-22-02296-f010:**
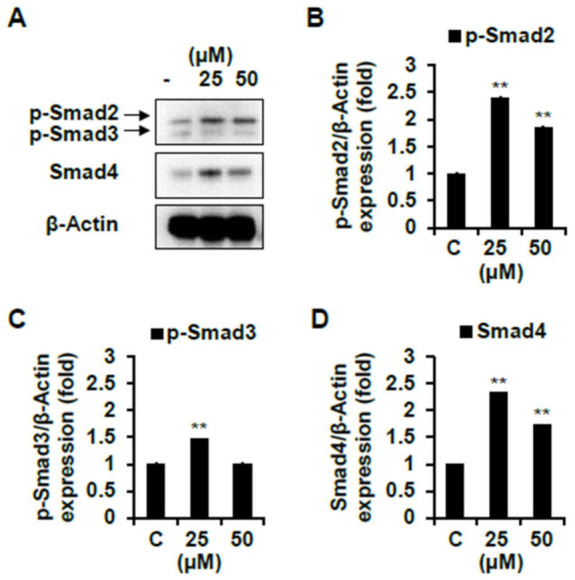
Skeletal muscle cell differentiation activity of compound **7** on C2C12 differentiated cells. The levels of p-Smad2/3 and Smad4 proteins were determined using a Western blot analysis. Every level of proteins versus β-Actin were measured by densitometry. C, control. Experiments were performed in triplicate and the data were expressed as mean ± SE; ** *p* < 0.01 as compared to the control (untreated cells).

**Figure 11 ijms-22-02296-f011:**
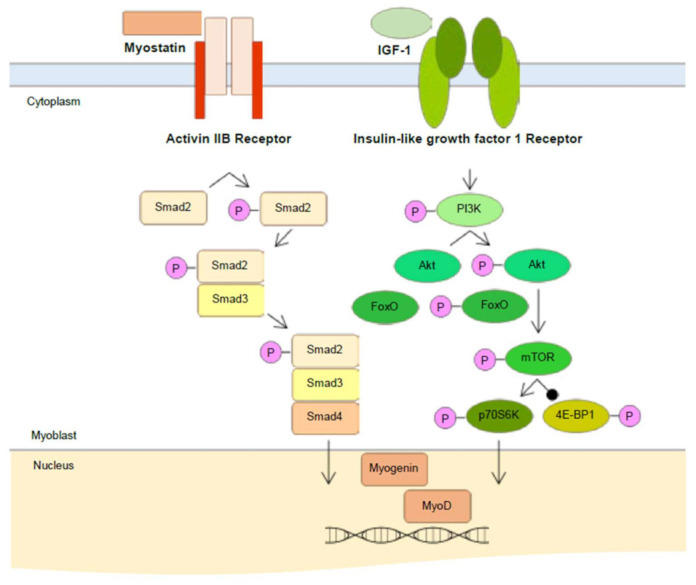
Mechanisms of Myostatin and IGF-1 mediated regulation of skeletal muscle differentiation.

**Table 1 ijms-22-02296-t001:** NMR spectroscopic data for compound **1.**

Pos.	*δ* _C_ ^a,b^		*δ*_H_^a,c^ (*J* in Hz)	Pos.	*δ* _C_ ^a,b^		*δ*_H_^a,c^ (*J* in Hz)
Kaempferol				Glucose B			
2	158.5			1′′′	106.2		4.75 (d, 7.5)
3	135.3			2′′′	75.7		3.66 *
4	180.0			3′′′	77.9		3.49 *
5	163.1			4′′′	72.0		3.41 *
6	99.9		6.12 (d, 2.1)	5′′′	76.3		3.43 *
7	165.7			6′′′	62.4		3.64 *, 3.51 *
8	94.8		6.18 (d, 2.1)				
9	158.3			Ferulic acid			
10	105.8			1′′′′	127.4		
				2′′′′	111.2		6.82 (d, 1.9)
1’	122.6			3′′′′	149.2		
2’	132.8		8.06 (m)	4′′′′	150.5		
3’	116.4		6.90 (m)	5′′′′	116.3		6.65 (d, 8.1)
4’	161.7			6′′′′	124.0		6.72 (dd, 1.9, 8.2)
5’	116.4			α	146.9		7.34 (d, 15.9)
6’	132.8		-	β	115.0		6.05 (d, 15.9)
				-COO	169.0		
Glucose A				-OCH_3_	56.3		3.79 (s)
1’’	101.3		5.12 (d, 7.5)				
2’’	84.7		3.71 *				
3’’	78.3		3.11 *				
4’’	71.0		3.39 *				
5’’	77.8		3.53 *				
6’’	64.8		4.44 *, 4.45 *				

^a^ Measured in methanol-*d_4_*, ^b^ 100 MHz, ^c^ 400 MHz, * overlapped signal, assignments were done by HSQC, HMBC, COSY, and NOESY experiments.

## Data Availability

The data presented in this study are available on request from the corresponding author.

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
