# Peer review of "Phytochemicals in Chinese Chive (Allium tuberosum) Induce the Skeletal Muscle Cell Proliferation via PI3K/Akt/mTOR and Smad Pathways in C2C12 Cells"

_ijms, 2021, doi:10.3390/ijms22052296_

Round 1

Reviewer 1 Report

In this manuscript, the authors have isolated phytochemicals from Chinese Chives and identified compounds that have a proliferative effect on muscle cells. They performed characterization to identify the compounds in the plant extract and further identified compounds from the mixture of chemicals that had a positive effect on muscle cell growth and differentiation. There are some points noted below that need to be addressed prior to acceptance. I would recommend reconsideration upon revision.

  1. Given the report where the Chinese Chives were found to be cytotoxic and have anti-proliferative effect, why did the authors think to test for increased proliferation in muscle cells? Are there any other reports that point to the proliferative effects of Chinese Chives? The authors must elaborate their rationale for choosing to explore the proliferative activity of Chinese Chives.
  2. Page 6: The authors mention “However, other five compounds did not exert any effect on cell proliferation.” However, compounds 3,4 and 5 appear to reduce cell proliferation. The authors should comment on this observation. Was any statistical analysis performed to determine whether these differences for compounds 3,4 and 5 are significant? If there were any statistical analyses performed to show that these differences in compounds 3,4 and 5 were not significant, that should be mentioned on the plot or figure legend.
  3. Page 7: The authors mention “Collectively, both compounds induce muscle cell growth through p-mTOR/p70S6K via the p-PI3K/Akt, and compound 7, especially, is believed to activate this signaling pathways more than compound 1”. Is this difference statistically significant? The authors should perform statistical analysis to make this claim.
  4. Page 7: The authors mention “As shown in Figure 7G and 8G, the protein expression level of MyoD was significantly increased by both compounds 1 and 7. However, the expression level of MyoD was higher in the cells treated with compound 1 than that of 7 regardless of the concentrations”. Similar to the comments for figures 7 and 8 E-F, the authors should perform statistical analysis to ensure that the difference is statistically significant to make this claim.
  5. Why was compound 2 not tested even though it had activity comparable to 1 and 7?
  6. The authors should perform some synergy experiments- for example, mix of 1&7 or 2&7, 1&2 or 1,2 and 7 to see if there was any improvement in the observed potency by either more upregulation of PI3K/Akt/mTOR or more downregulation of Smad pathways or a combination of both effects compared to the compounds used in isolation.

Author Response

Reviewer 1

In this manuscript, the authors have isolated phytochemicals from Chinese Chives and identified compounds that have a proliferative effect on muscle cells. They performed characterization to identify the compounds in the plant extract and further identified compounds from the mixture of chemicals that had a positive effect on muscle cell growth and differentiation. There are some points noted below that need to be addressed prior to acceptance. I would recommend reconsideration upon revision.

  1. Given the report where the Chinese Chives were found to be cytotoxic and have anti-proliferative effect, why did the authors think to test for increased proliferation in muscle cells? Are there any other reports that point to the proliferative effects of Chinese Chives? The authors must elaborate their rationale for choosing to explore the proliferative activity of Chinese Chives.

Answer: There are not many reports on skeletal muscle cell proliferation effect of natural products yet, but some reports of Allium genus related to muscle growth have been reported. Gautam et al (2015) confirmed that onion stimulates glucose uptake by the rat skeletal muscle cells (L6). Glucose uptake is related to skeletal muscle growth and mass increase (Yang, 2014). Also, macrostemonoside A, a new compound isolated from A. macrostemon Bung, is supposed to promote energy metabolism in muscles during its action mechanisms on mice (Xie et al., 2008). More recently, Kalhotra et al. (2020) treated garlic bulb extract on L6 cells and observed the percent of cell proliferation corresponding to differentiated skeletal muscle cells compared to untreated cells. Based on those several reports of the Allium genus, we selected Chinese chives as our sample, whose muscle growth activity was still unknown. We added the above results of Allium genus in revised manuscript.

  1. Page 6: The authors mention “However, other five compounds did not exert any effect on cell proliferation.” However, compounds 3,4 and 5 appear to reduce cell proliferation. The authors should comment on this observation. Was any statistical analysis performed to determine whether these differences for compounds 3,4 and 5 are significant? If there were any statistical analyses performed to show that these differences in compounds 3,4 and 5 were not significant, that should be mentioned on the plot or figure legend.

Answer: Thank you for your comment. We reviewed the sentence you mentioned. And, we explained additionally the results of the treatment of compound 3, 4, and 5. The added sentences are highlighted in red color and you can find it in the Results section 2.3. In addition, since Figure 6B is a result of cell proliferation, statistical results for compound 3, 4, and 5 that decreased cell proliferation were not added. However, the experimental group treated with all compounds including compound 3, 4, and 5 was statistically compared with the control group as described in the legend Figure 6.

  1. Page 7: The authors mention “Collectively, both compounds induce muscle cell growth through p-mTOR/p70S6K via the p-PI3K/Akt, and compound 7, especially, is believed to activate this signaling pathways more than compound 1”. Is this difference statistically significant? The authors should perform statistical analysis to make this claim.

Answer: We appreciate your comment. We noticed that there was an error in the sentence you mentioned. The expressions of p-PI3K, p-Akt, p-mTOR, and p-p70S6K were all increased in the cells treated with compound 1 and 7 compared to the control group. However, we described that compound 7 induces muscle cell growth through p-mTOR/p70S6K via p-PI3K/Akt rather than compound 1 based on simple numbers. Therefore, to clarify this result, we analyzed statistical significance. Data were statistically compared by two-tailed one-way ANOVA and Tukey’s post-test using Prism software (Version 4.00; GraphPad Inc., La Jolla, CA, USA). Data were considered statistically significant at p < 0.05). As a result, as can be seen from supplementary data, compound 7 increased protein expression levels of p-PI3K, p-Akt, p-mTOR, and p-p70S6K protein than compound 1.

  1. Page 7: The authors mention “As shown in Figure 7G and 8G, the protein expression level of MyoD was significantly increased by both compounds 1 and 7. However, the expression level of MyoD was higher in the cells treated with compound 1 than that of 7 regardless of the concentrations”. Similar to the comments for figures 7 and 8 E-F, the authors should perform statistical analysis to ensure that the difference is statistically significant to make this claim.

Answer: Thank you for your comment. We noticed the same error in the description of the results of MyoD protein expressions. Therefore, as your advice, we analyzed the statistical significance of Figure 7 and 8. As a result, as shown in supplementary data, compound 7 increased protein expression levels of MyoD protein than compound 1.

  1. Why was compound 2 not tested even though it had activity comparable to 1 and 7?

Answer: We selected compound 7, which had the highest skeletal muscle cell proliferation effect, as a potential effect substance for muscle growth, and analyzed mechanisms such as PI3K/Akt/mTOR and Smad pathways. And, since compound 7 was a newly discovered compound in the present study and showed the effect of muscle cell proliferation. Therefore, the mechanism analysis was performed by selecting compound 1 as the second potential effective substance for muscle cell growth. In the case of compound 2, this is the flavonoid glycoside we already know. This compound 2 is also considered a potential substance for muscle cell growth because it has a similar chemical structure to compound 1 and plans to analyze the mechanism.

  1. The authors should perform some synergy experiments- for example, mix of 1&7 or 2&7, 1&2 or 1,2 and 7 to see if there was any improvement in the observed potency by either more upregulation of PI3K/Akt/mTOR or more downregulation of Smad pathways or a combination of both effects compared to the compounds used in isolation.

Answer: thank you for suggesting the synergy experiments to see the improvement in the observed potency by combination effects. However, in this study, we wanted to see the skeletal muscle cell proliferation activity of each of the active ingredients isolated from Chinese chives. In terms of natural products, we focused on the fact that ‘various ingredients of Chinese chives have the muscle proliferation activity’ rather than ‘the synergistic effect that can maximize the muscle proliferation activity’. We will apply the reviewer’s suggestion to further study.

Reviewer 2 Report

The authors in their research article describe the isolation of phytochemicals from Allium tuberosum and the evaluation of their ability to induce C2C12 cell proliferation. Among the isolated compounds, kaempferol-3-O-(6''-feruloyl)-sophoroside and 5-aminouridine were found to induce the highest proliferation of C2C12 cells. Additionally, these compounds influenced muscle-related signaling pathways by up-regulating the PI3K/Akt/mTOR pathway and downregulating the Smad pathway.

Overall, the manuscript presents novel findings regarding the biological activity of Allium tuberosum phytochemicals and their effects on skeletal muscle proliferation. I have the following questions for the authors regarding their study:

The induction of cell proliferation, shown with the 5- bromo-2'-deoxyuridine (BrdU) proliferation assay, is not that prominent. What was the incubation period of cells in this assay? The authors could examine proliferation at different time periods, extending the period of  incubation with the phytochemicals. Furthermore, the authors should provide information regarding the incubation time of cells below Figure 6 as well as in the Materials and Methods Section. 

Please specify the density of cells seeded on plates used in the cytotoxicity assay and proliferation assay in the Materials and Methods Section. What was the cell density used to determine cell proliferation. Higher densities stimulate proliferation, therefore, this assay should also be verified at lower cell densities.

Please specify the amount of serum supplemented in the medium used in the proliferation assay. Since serum stimulates cell proliferation, the authors should perform this assay also with reduced serum levels.

In the Western blot assay please specify the dilutions of antibodies used.

Since the authors did not examine the effects of the Chinese chive extract on cell proliferation I would suggest changing the conclusion that the Chinese chive might be used as an aid in stimulating muscle growth and differentiation. The authors in this study examined the activity of  isolated compounds, through interactions between the extract constituents the extract could have a different effect.

Author Response

Reviewer 2

The authors in their research article describe the isolation of phytochemicals from Allium tuberosum and the evaluation of their ability to induce C2C12 cell proliferation. Among the isolated compounds, kaempferol-3-O-(6''-feruloyl)-sophoroside and 5-aminouridine were found to induce the highest proliferation of C2C12 cells. Additionally, these compounds influenced muscle-related signaling pathways by up-regulating the PI3K/Akt/mTOR pathway and downregulating the Smad pathway.

Overall, the manuscript presents novel findings regarding the biological activity of Allium tuberosum phytochemicals and their effects on skeletal muscle proliferation. I have the following questions for the authors regarding their study:

  1. The induction of cell proliferation, shown with the 5- bromo-2'-deoxyuridine (BrdU) proliferation assay, is not that prominent. What was the incubation period of cells in this assay? The authors could examine proliferation at different time periods, extending the period of incubation with the phytochemicals. Furthermore, the authors should provide information regarding the incubation time of cells below Figure 6 as well as in the Materials and Methods Section.

Answer: As shown in Figure 6B, among the seven phytochemicals, cells treated with compound 1, 2, and 7 significantly proliferated compared to control cells. And, the cell culture period was differentiated for a total of 6 days as described in Materials and Method section 3.8. However, in order to accurately describe the processing time of the sample, we have added information at the end of materials and Method section 3.9. And the added part is highlighted in red color. In addition, cells incubation time information for measuring cell toxicity and cell proliferation activity has been added to the legend in Figure 6 and also is highlighted in red color.

And in this study, we measured whether cell proliferation is activated by phytochemicals treatment. Therefore, according to the method suggested in the previous research papers, we set the differentiation period from myoblast to myocyte to six days, and during this period, we measured how active cell proliferation is by treating phytochemicals. Therefore, phytochemicals were not treated at other times or analyzed by extending the incubation period.

* Previous research papers

  1. [32] Muthuramalingam, K.; Kim, S.-Y.; Kim, Y.; Kim, H.-S.; Jeon, Y.-J.; Cho, M. Bigbelly seahorse (Hippocampus abdominalis)-derived peptides enhance skeletal muscle differentiation and endurance performance via activated P38MAPK/AKT signalling pathway: An in vitro and in vivo analysis. J. Funct. Foods 2019, 52, 147–155.
  2. [33] Kim, S.-Y.; Kim, H.-S.; Cho, M.; Jeon, Y.-J. Enzymatic Hydrolysates of Hippocampus abdominalis Regulates the Skeletal Muscle Growth in C2C12 Cells and Zebrafish Model. J. Aquat. Food Prod. Technol. 2019, 28 (3), 264–274

  1. Please specify the density of cells seeded on plates used in the cytotoxicity assay and proliferation assay in the Materials and Methods Section. What was the cell density used to determine cell proliferation. Higher densities stimulate proliferation, therefore, this assay should also be verified at lower cell densities.

Answer: Response: As your recommend, we further described the density of cells seeded on plates used for cytotoxicity and proliferation assays in Materials and Method section 3.9. For cytotoxicity, cells were seeded at 1 × 105 cells per well (96 well plate), and after 24 h of sample treatment, the cytotoxicity was assessed using MTT assay. Cells were seed at 5 × 104 cells per well (96 well plate), and the phytochemicals were treated simultaneously with three medium replacements during the differentiation period. After, the cell proliferation activity was assessed using BrdU assay. This addition can be found in Materials and Method section 3.9. and is highlighted in red color.

And in your opinion, it is true that the higher the density of cells, the more proliferation is promoted. However, in the present study, we tried to analyze whether cell proliferation is activated by the treatment of phytochemicals along with the differentiation induction compared to the cells that induced only differentiation without any treatment. Therefore, we seeded the cells at 5 × 104 cells per well to measure the cell proliferation activity based on the method presented in the previous research paper, and measured the cell proliferation activity by phytochemicals under this condition.

* Previous research papers

  1. [32] Muthuramalingam, K.; Kim, S.-Y.; Kim, Y.; Kim, H.-S.; Jeon, Y.-J.; Cho, M. Bigbelly seahorse (Hippocampus abdominalis)-derived peptides enhance skeletal muscle differentiation and endurance performance via activated P38MAPK/AKT signalling pathway: An in vitro and in vivo analysis. J. Funct. Foods 2019, 52, 147–155.
  2. [33] Kim, S.-Y.; Kim, H.-S.; Cho, M.; Jeon, Y.-J. Enzymatic Hydrolysates of Hippocampus abdominalis Regulates the Skeletal Muscle Growth in C2C12 Cells and Zebrafish Model. J. Aquat. Food Prod. Technol. 2019, 28 (3), 264–274.

  1. Please specify the amount of serum supplemented in the medium used in the proliferation assay. Since serum stimulates cell proliferation, the authors should perform this assay also with reduced serum levels.

Answer: It is correct that serum stimulates cell proliferation. However, we used DMEM medium containing 2% horse serum, a method suggested in previous research papers, as the differentiation medium to induce differentiation from myoblasts to myocytes. And compared with the control group treated with only differentiation medium, it was verified whether treatment with phytochemicals stimulates cell proliferation. As a result, compared to the control group, the cells treated with compound 1, 2, and 7 were activated to proliferate. Therefore, it is believed that we do not need to analyze cell proliferation with reduced serum.

* Previous research papers

  1. [32] Muthuramalingam, K.; Kim, S.-Y.; Kim, Y.; Kim, H.-S.; Jeon, Y.-J.; Cho, M. Bigbelly seahorse (Hippocampus abdominalis)-derived peptides enhance skeletal muscle differentiation and endurance performance via activated P38MAPK/AKT signalling pathway: An in vitro and in vivo analysis. J. Funct. Foods 2019, 52, 147–155.
  2. [33] Kim, S.-Y.; Kim, H.-S.; Cho, M.; Jeon, Y.-J. Enzymatic Hydrolysates of Hippocampus abdominalis Regulates the Skeletal Muscle Growth in C2C12 Cells and Zebrafish Model. J. Aquat. Food Prod. Technol. 2019, 28 (3), 264–274.

  1. In the Western blot assay please specify the dilutions of antibodies used.

Answer: Thank you for your recommend. We described the dilution ratio of each primary and secondary antibodies used for Western blot analysis in method section 3.10. and are highlighted in red color.

  1. Since the authors did not examine the effects of the Chinese chive extract on cell proliferation I would suggest changing the conclusion that the Chinese chive might be used as an aid in stimulating muscle growth and differentiation. The authors in this study examined the activity of isolated compounds, through interactions between the extract constituents the extract could have a different effect.

Answer: Thank you for the suggestion. As reviewer commented, the muscle cell proliferation effects of extracts and the extract constituents is different. So we changed the conclusion sentence to ‘Taken together, we suggest that the major constituents of Chinese chive, flavonoids and amino acids, might be used as functional food candidate for aiding muscle growth and differentiation.’ in revised manuscript.

Round 2

Reviewer 1 Report

In the revised version, the authors have addressed most of the concerns raised earlier. A couple of minor points are noted below.

  1. In the rebuttal letter, under point #4, the authors state "As a result, as shown in supplementary data, compound 7 increased protein expression levels of MyoD protein than compound 1". However, in the main text of the manuscript, on page 7, last line, the sentence still says "However, the expression level of MyoD was higher in the cells treated with compound 1 than that of 7 regardless of the concentrations (Figure S7)". The authors must correct this discrepancy.
  2. In figure S7, please clarify what 'a', 'b', 'c' etc. mean on the bar graphs. This information can be added to the figure legend. 

Author Response

  1. In the rebuttal letter, under point #4, the authors state "As a result, as shown in supplementary data, compound 7 increased protein expression levels of MyoD protein than compound 1". However, in the main text of the manuscript, on page 7, last line, the sentence still says "However, the expression level of MyoD was higher in the cells treated with compound 1 than that of 7 regardless of the concentrations (Figure S7)". The authors must correct this discrepancy.

Answer: We noticed the error in the description of the results of MyoD protein expressions in the main text of the manuscript. Therefore, we revised the description of the results of MyoD protein expression as follows: “The expression level of MyoD protein was also higher in the cells treated with compound 7 than that of 1 (Figure S7). In addition, it is believed to be due to the results that the expression level of PI3K/Akt/mTOR was more activated by treating compound 7 than that of compound 1.”

  1. In figure S7, please clarify what 'a', 'b', 'c' etc. mean on the bar graphs. This information can be added to the figure legend. 

Answer: Thank you for your comment. a-eValues having different superscripts are significantly different at p < 0.05. We added this significance information to the figure legend for figure S7.

Reviewer 2 Report

The authors have addressed my comments and I have no further questions. 

Author Response

(The authors gave the same response as above.)
